Integrating single-cell and bulk sequencing data to identify glycosylation-based genes in non-alcoholic fatty liver disease-associated hepatocellular carcinoma

Zhou Zhijia 1
Gao Yanan 2
Deng Longxin 2
Lu Xiaole 2
Lai Yancheng 2
Wu Jieke 2
Chen Shaodong 3
Li Chengzhong leo_lee66@126.com 4
Liang Huiqing 13850005898@163.com 5 6
1 Department of Hepatology, ShuGuang Hospital Affiliated to Shanghai University of Traditional Chinese Medicine , Shanghai , China
2 The First School of Clinical Medicine, Southern Medical University , Guangzhou , Guangdong Province , China
3 Xiamen University , Xiamen , Fujian Province , China
4 Changhai Hospital, The Second Military Medical University , Shanghai , China
5 Hepatology Unit, Xiamen Hospital of Traditional Chinese Medicine , Xiamen , Fujian Province , China
6 College of Traditional Chinese Medicine, Beijing University of Traditional Chinese Medicine , Beijing , China
Banerjee Priyanka
Electronic publication date: 2024 Mar 18
Publication date: 2024
Volume: 12
Electronic Location ID: e17002
Received 2023 Jul 28; Accepted 2024 Feb 5
Copyright: ©2024 Zhou et al.
Copyright year: 2024
Copyright holder: Zhou et al.
License: This is an open access article distributed under the terms of the Creative Commons Attribution License, which permits unrestricted use, distribution, reproduction and adaptation in any medium and for any purpose provided that it is properly attributed. For attribution, the original author(s), title, publication source (PeerJ) and either DOI or URL of the article must be cited.
License URL: https://creativecommons.org/licenses/by/4.0/

Keywords: Glycosylation, Nonalcoholic fatty liver disease, Hepatocellular carcinoma, Immunotherapy, Machine learning

Funding: National Science Natural Foundation of China 82174141 82374353 This work was supported by the National Science Natural Foundation of China (Nos. 82174141 and 82374353). The funders had a role in study design, data collection and analysis, decision to publish, or preparation of the manuscript.

==============================
Background

The incidence of non-alcoholic fatty liver disease (NAFLD) associated hepatocellular carcinoma (HCC) has been increasing. However, the role of glycosylation, an important modification that alters cellular differentiation and immune regulation, in the progression of NAFLD to HCC is rare.

Methods

We used the NAFLD-HCC single-cell dataset to identify variation in the expression of glycosylation patterns between different cells and used the HCC bulk dataset to establish a link between these variations and the prognosis of HCC patients. Then, machine learning algorithms were used to identify those glycosylation-related signatures with prognostic significance and to construct a model for predicting the prognosis of HCC patients. Moreover, it was validated in high-fat diet-induced mice and clinical cohorts.

Results

The NAFLD-HCC Glycogene Risk Model (NHGRM) signature included the following genes: SPP1, SOCS2, SAPCD2, S100A9, RAMP3, and CSAD. The higher NHGRM scores were associated with a poorer prognosis, stronger immune-related features, immune cell infiltration and immunity scores. Animal experiments, external and clinical cohorts confirmed the expression of these genes.

Conclusion

The genetic signature we identified may serve as a potential indicator of survival in patients with NAFLD-HCC and provide new perspectives for elucidating the role of glycosylation-related signatures in this pathologic process.

Introduction

Over the past few years, there has been a notable increase in the prevalence of non-alcoholic fatty liver disease (NAFLD), with an estimated global prevalence of approximately 32.4%, compared to 25.5% in or before 2005 (Riazi et al., 2022). NAFLD can advance from simple fatty liver to steatohepatitis, liver fibrosis, cirrhosis, and potentially to hepatocellular carcinoma (HCC) (Huang et al., 2023). Non-alcoholic steatohepatitis (NASH), an advanced type of NAFLD, is associated with an estimated annual incidence rate of HCC ranging from 0.5% to 2.6% in patients with NASH-induced cirrhosis (Huang et al., 2023; Dolgin, 2023). Although the incidence rate of HCC linked to NAFLD is lower than that of HCC linked to other causes, such as hepatitis C, the large number of people affected by NAFLD still signifies a substantial population potentially at risk (Huang, El-Serag & Loomba, 2021).

NAFLD is typically asymptomatic, and it is currently not advised to conduct regular screening, even for high-risk patients. Therefore, NAFLD is often not intervened until symptoms occur, making it less predictable and preventable (Westfall & Jeske, 2020). Although noninvasive tests are useful for ruling out advanced liver disease in patients, histological assessment remains the preferred approach for diagnosing, prognosing, and monitoring the treatment of NAFLD; however, concerns exist regarding its accuracy and pathologist expertise (Li et al., 2022; Paul, 2020; Kechagias et al., 2022). Hence, it is essential to uncover a novel and precise approach to diagnose NAFLD, address the limitations of pathological diagnosis, and identify potential therapeutic targets.

Glycosylation, an essential protein modification process, can affect cell differentiation, tumor progression, and immune regulation (Ohtsubo & Marth, 2006; Josic, Martinovic & Pavelic, 2019). Recent studies have highlighted its significant role in HCC progression by modulating pro-tumor signaling pathways and altering protein function. For instance, excessive O-GlcNAcylation can enhance hepatocyte malignancy (Zhu et al., 2012), and the specific N-glycosylation of MerTK can promote HCC growth (Liu et al., 2022). Thus, abnormal glycosylation, which is indicative of HCC protein alterations, could be a potential diagnostic and prognostic biomarker. Given the unresolved challenges in diagnosing and treating NAFLD, it is crucial to continue exploring the genes associated with NAFLD glycosylation to gain new insights into prognosis and treatment strategies.

In this study, we focused on glycosylation-related genes (GRGs) in NAFLD by analyzing single-cell data from patients with NAFLD-related HCC obtained from the GSE189175 dataset. Differentially expressed genes (DEGs) were identified in cells exhibiting high and low levels of expression of GRGs. To further investigate the implications of these DEGs, unsupervised cluster analysis was performed on patients with HCC using the Cancer Genome Atlas Liver Hepatocellular Carcinoma (TCGA-LIHC) dataset, resulting in the classification of patients into two distinct classes and the identification of differential genes. Using weighted gene co-expression network analysis (WGCNA), candidate genes associated with prognosis and glycosylation were identified. Subsequently, a comprehensive approach involving univariate Cox regression, Least Absolute Shrinkage and Selection Operator (LASSO), and multi-way Cox regression was employed to select six key genes for constructing a diagnostic model, namely the NAFLD-HCC GlycoGene Risk Model (NHGRM). These essential genes present potential as molecular indicators and treatment objectives for NAFLD, aiding a more thorough understanding of the function of glycosylation in the development of NAFLD.

Overall, the results of this study offer new perspectives on NAFLD molecular markers and potential therapeutic targets, advancing our knowledge of glycosylation in NAFLD progression.

Materials & Methods

Data retrieving of single cell and bulk sequencing data

The Single-Cell RNA Sequencing (scRNA-seq) data for HCC (accession number: GSE189175) (Rao et al., 2021) were sourced from the Gene Expression Omnibus (GEO) database (https://www.ncbi.nlm.nih.gov/geo/). The dataset comprised tumor and non-tumor cells from three paraneoplastic and tumor tissues, totaling 59,915 cells. Subsequently, we obtained the TCGA-LIHC dataset from The Cancer Genome Atlas (https://portal.gdc.cancer.gov/). The dataset comprised of 50 normal samples and 371 tumor samples containing RNA sequence data. To streamline further analysis, the RNA sequence data from TCGA dataset were converted from fragments per kilobyte per million (FPKM) to transcripts per million reads (TPM).

A comprehensive set of 636 GRGs was downloaded from the Molecular Signature Database (MSigDB, http://www.gsea-msigdb.org/gsea/msigdb/) (Liberzon et al., 2015) (Table S1), a web-based repository of annotated gene sets for biofunctional analysis. To investigate the tumor microenvironment (TME), tumor immune infiltration was estimated using ESTIMATE and CIBERSORT (Becht et al., 2016) methods.

Apart from the previously mentioned datasets, datasets associated with NAFLD, such as GSE54236 (Villa et al., 2016), GSE89632 (Arendt et al., 2015) and GSE48452 (Ahrens et al., 2013) were downloaded from the GEO database. Gene expression profile data and the corresponding annotation files were obtained. For RNA-seq data, gene expression was quantified and normalized using the “normalizebetweenarrays” function in the “limma” package, and log2 transformations were applied.

Process for single cell sequencing

The scRNA-seq data were subjected to quality control (QC) using the R packages “seurat” (Butler et al., 2018) and “singleR” (Aran et al., 2019). To ensure data accuracy, genes that manifest expression in a population of less than three individual cells, cells containing a gene count below 200 or in excess of 7,000, and those with mitochondrial gene content surpassing 5% were excluded from the analysis. Following these filters, a total of 318,234 cells were selected for subsequent analysis. The selected cells were then subjected to scaling and normalization using a linear regression model with log normalization.

To identify genes with high variability, we employed the “FindVariableFeatures” function from the Seurat package and identified the top 3,000 genes that exhibited significant variability. To mitigate any potential batch effects that might influence subsequent analyses, the harmony package was utilized for batch-effect normalization across the specimens. By assessing the top 20 principal components (PCs) using the t-distributed Stochastic Neighbor Embedding (t-SNE) algorithm, we were able to visualize the formation of distinct clusters within the dataset. The “FindNeighbors” and “FindClusters” functions were then utilized with the resolution parameter set to 1, resulting in the identification of 23 distinct cell clusters.

To ascertain the DEGs within each cluster, we leveraged the “FindAllMarkers” function from “seurat” package. The “singleR” package was employed to annotate the cell types based on the characteristic markersof the identified clusters. The annotations were manually validated against published literature.

AddModuleScore

We assigned a glycosylation activity score (G-score) to each cell using the AddModuleScore function in the Seurat R package. Principally, the row-means function was employed to calculate the mean expression values of all genes. Based on this mean, the expression matrix was divided into multiple bins according to the specified number set using the nbin parameter. After setting a random number of seeds, 100 background genes were randomly selected from each bin in the matrix. Next, we computed the gene expression ranking for each cell by using the area under the curve (AUC) value of the chosen GRGs. This approach facilitated the evaluation of the proportion of gene sets manifesting elevated expression, wherein cells possessing augmented AUC values signified enhanced magnitudes of gene transcription. Subsequently, we divided all the cells into high G-score and low G-score groups according to the median G-score.

Gene set variation analysis

To explore the enriched biological pathways in different G-score subgroups, we conducted a gene set variation analysis (GSVA) using the GSVA R package. The results of this analysis are presented as bar graphs illustrating all the significantly different pathways.

Unsupervised clustering analysis

Differential gene analysis was conducted between the high and low G-score groups in the single-cell data using the FindMarkers function with designated criteria (min.pct = 0.25, logfc.threshold = 0.25). A total of 193 genes with an adjusted P-value (p.adj) below 0.05, were identified as significant.

Thus these 193 genes were subsequently utilized to construct consensus clusters and determine subtypes within the TCGA-LIHC dataset using the ConsensusClusterPlus (Wilkerson & Hayes, 2010) R package. The partitioning around the median (PAM) algorithm was employed, and distances were measured using “1-Pearson” correlation coefficients. To ensure robustness, 100 bootstrap replicates were performed, randomly selecting 80% of the patients from the GEO cohort. The k-values (number of clusters) used for clustering ranged from two to nine. Finally, a Pearson correlation analysis was performed to assess the associations among essential genes and 193 previously identified genes.

Gene set enrichment analysis

To assess glycosylation scores in each participant in the TCGA-LIHC cohort, we performed single-sample gene set enrichment analysis (ssGSEA). Participants with high glycosylation were distinguished from those with low glycosylation.

Weighted gene co-expression network analysis

We employed weighted gene co-expression network analysis (WGCNA) (Langfelder & Horvath, 2008) to study the gene co-expression networks in the TCGA-LIHC dataset. The following steps were performed: (1) The ‘goodSamplesGenes’ function is used to filter out genes with missing data; (2) Tumor samples are classified and outliers are removed; (3) A cut line (minimum module size of 100) is implemented; (4) Ascertaining the optimal soft threshold to compute the adjacency matrix through a graphical approach; (5) Generation of an adjacency matrix to assess genetic interconnectivity within the network; (6) Using the adjacency matrix as a starting point, build a topological overlap matrix (TOM); (7) Conduct a hierarchical clustering using the average linkage approach while taking TOM differences into account; (8) Dynamically trimming the dendrogram to pinpoint modules characterized by elevated correlation coefficients (r > 0.25) and congruent transcriptional patterns; (9) Utilization of Pearson’s correlation assay to investigate relationships between eigengenes (epitomizing module transcriptional profiles) and clinical attributes; (10) Isolation of modules that manifest strong associations with clinical parameters, encompassing glycosylation index, vitality status, and longevity of survival, for ensuing in-depth analysis.

Constructing risk scoring method

We performed differential gene expression analysis to compare different glycosylation groups in the scRNA-seq data of NAFLD-associated HCC patients. The primary objective was to identify the gene modules linked to glycosylation and survival outcomes using WGCNA. Subsequently, univariate analysis was employed to select genes that exhibited statistically significant correlations with overall survival (OS) of patients (P <0.001). To refine the gene set and establish robust prognostic associations, LASSO analysis was applied, resulting in the identification of risk coefficients. The “glmnet” software tool was used to build a risk model that assigned a risk score based on the LASSO coefficients to each HCC patient in the TCGA-LIHC dataset. The patients were separated into NHGRM_low and NHGRM_high groups using the median risk score as a criterion. The Kaplan–Meier (K-M) method was used to generate prognostic survival curves. The accuracy of the prediction results was further validated using an independent dataset (GSE54236) (Villa et al., 2016) through survival analysis and the computation of AUC values.

Methods for assessing the independence and validity of predictive models

We developed a composite nomogram with parameters combining NHGRM, age, TMN stage, grade, and gender to predict the OS at 1, 3, and 5 years in patients with HCC. Using calibration curves and receiver operating characteristic (ROC) curves, the precision of this nomogram was evaluated using calibration and receiver operating characteristic curves. To determine its net benefit, decision curve analysis (DCA) was conducted. To investigate the prognostic importance of the NHGRM, subgroup analyses were conducted to assess its prognostic value in various clinical subgroups, including age, gender, and clinical stage.

Explorating the association between prognostic models and neoplastic immunological features and their influence on immune-modulating therapeutics

We used the CIBERSORT R package to assess the relative proportions of different cell types in different NHGRM groups. This software package helps assess the components present in the TME, with higher scores indicating a higher abundance of these components. In addition, certain immune cells express molecules called immune checkpoints that modulate immunological activation and curb hyperactive immune reactions. The expression profiles of an array of established immune checkpoint genes (ICG) derived from the scholarly corpus were meticulously examined across two disparate populations. This assessment sought to substantiate the model’s competency in forecasting responses to immunotherapy.

Animals and experimental procedure

C57BL/6J mice were obtained from Shanghai Jiesijie Laboratory Animal Co., Ltd. (Shanghai, China) and housed under controlled conditions (23 ± 2 °C temperature, 60 ± 10% humidity, a 12-hour light-dark cycle, and free access to food and water). All animal experiments were approved by the Animal Experimental Ethics Committee of the Xiamen University (approval no. XMULAC20200055). After one week of acclimatization, the mice were randomly divided into two groups: a high-fat diet (HFD) model group and a normal diet control group (C group), with six mice in each group. In order to induce NAFLD mouse model according to previous method (Van Herck, Vonghia & Francque, 2017), mice were given a high-fat diet which consisting of 77.5% regular feed, 0.5% sodium cholate, 2% cholesterol, 5% soybean, 5% sucrose, and 10% lard for 16 weeks to induce a NAFLD mouse model. The mice in the control group were fed a normal diet. After 16 weeks, euthanasia of the mice was carried out by cervical dislocation according to the AVMA Guidelines on Euthanasia, and liver tissue samples were weighed and collected. Specifically, the mice were administered an intraperitoneal injection of 1% sodium pentobarbital solution at a dose of 80 mg/kg before being sacrificed for cervical dislocation. The thumb and index finger were placed on either side of the neck, at the base of the mouse skull. In contrast, the base of the tail or hind limbs is quickly pulled, causing separation of the cervical vertebrae from the skull. All mice were sampled and included in subsequent analyses. Liver tissue was used for quantitative Polymerase Chain Reaction(qPCR) and kept in RNase-free tubes before RNA extraction. The liver tissue used for the sections was rinsed with ice-cold PBS before hematoxylin–eosin staining.

RT-PCR

TRIzol reagent (Invitrogen, Carlsbad, CA, USA) was employed to extract total RNA from in vitro cultured cells, followed by cDNA synthesis using a Reverse Transcription Master kit (Invitrogen, Carlsbad, CA, USA). Subsequently, TB Green Premix Ex Taq II (TaKaRa, Shiga, Japan) was used to conduct quantitative real-time PCR (qRT-PCR). The primer sequences used in this study are listed in Table S4. Relative gene expression was evaluated using the comparative 2(−ΔΔCT) method, with mRNA levels normalized to expression in normal mouse livers.

Hematoxylin–eosin staining

The tissues were fixed in 4% paraformaldehyde for at least 24 h and subsequently embedded in paraffin. Sections of 4 µm thickness were obtained using a Leica microtome. Hematoxylin and eosin (H&E) staining was performed to visualize the tissue structure. Prior to staining, lung, liver, spleen, and kidney tissues were treated with 4% paraformaldehyde. Tissue sections were embedded in paraffin and secured to ensure a consistent section thickness of 5 µm. Following a standardized protocol, all samples were stained with H&E staining.

ShuGuang cohort and immunofluorescence

Formalin-fixed paraffin-embedded (FFPE) liver tissue blocks from NAFLD-HCC patients at ShuGuang Hospital Affiliated to Shanghai University of Traditional Chinese Medicine under ShuGuang Hospital Institutional Review Board (IRB)-approved protocols (No. 2018-630-59-01), and all patients gave informed consent for the collection of clinical information, tissue collection, and study testing. For immunofluorescence of liver samples, sections of liver FFPE tissues were incubated with Osteopontin (OPN), S100A9 and RAMP3 separately for 30 min at 37 °C The nucleus was stained with DAPI (Table S5).

Statistical analysis

Frequencies and proportions were used to present categorical variables, whereas median (interquartile range) or mean (standard deviation) was used to show continuous variables. Using the K-M technique, the median overall survival was calculated along with 95% confidence intervals. The Kruskal–Wallis H-test was used to compare the statistical differences among the distinct cupro-clusters. The log-rank test was used to evaluate the differences in overall survival between the training and validation cohorts. The correlation coefficients between the components and the risk score were calculated using Spearman’s analysis. Except where otherwise noted, a P-value of 0.05 or less was considered statistically significant. All data analyses were performed using R version 4.1.2.

Results

scRNA profiling of NAFLD-HCC

As shown in Fig. S1, we aimed to identify glycosylated hub genes of prognostic significance in NAFLD-associated HCC by a combination of algorithms. Briefly, through four steps: (1) Using the Seurat R package’s AddModuleScore function, all cells were scored with G-scores based on 636 GRGs (Table S1), then divided into high and low G-score subgroups via median to identify differential genes in the scRNA-seq dataset; (2) Based on these differentially expressed genes, an unsupervised clustering algorithm was used to classify hepatocellular carcinoma patients in the TCGA database into two categories to verify that glycosylation levels can influence patient prognosis; (3) Using the GRGs as a reference, the ssGSEA algorithm was used to calculate the glycosylation level of each patient in the TCGA-LIHC, and then the WGCNA was used to screen for modules that were highly correlated with prognosis and glycosylation level; (4) then using univariate Cox-LASSO-multivariate Cox regression to screen the features among the modules and build a model that can predict the prognosis of HCC patients. Figure 1 provides a comprehensive view of the single-cell landscape of NAFLD-associated HCC tumors and paracancerous samples from the GSE189175 dataset. Subsequent categorization using the t-SNE method partitioned all cells into 23 distinct clusters contingent on the entire range of gene expression (resolution =1) (Fig. 1A). The triad of samples incorporated in this study demonstrated no discernible batch effects, as evidenced by the even distribution of cells within each individual specimen (Fig. 1B). In order to identify the major cellular subpopulations in the single cell data set, we used automated annotation combined with manual annotation to classify all cells into hepatocyte, myeloid cells, NK/T cells, cholangiocyte, and endothelial cells (Fig. 1C). Specifically, automated annotation was performed using the singerR R package, and manual annotation referenced a number of recognized cell markers (Fig. 1F and Fig. S2A). Figure 1D depicts the distribution of tumor and normal cells in the profiling of NAFLD-HCC. Based on the AddModuleScore function, all cells were assigned a score related to GRGs and were subsequently divided into subgroups with high and low G-scores through the median (Fig. 1E). FindMarkers fuction (min.pct = 0.25, logfc.threshold = 0.25, p_val_adj <0.05) in Seurat R package was used to obtain 193 differential expression genes between the high and low G-score groups(Table S1). In the present study, cells with high G-scores were predominantly found in hepatocytes, myeloid cells and endothelial cells (Fig. 1G). In an attempt to understand the probable biological mechanisms underlying these differences, we performed differential and functional examinations. To explore the biological pathways enriched in different G-score subgroups, the GSVA method was conducted, and it revealed that processes such as oxidative phosphorylation, adipogenesis, and fatty acid metabolism pathways were significantly prevalent in the high G-score group, as shown in Fig. 1H.

Figure 1 Single-cell landscape overview of GSE189175.

(A) Following quality assurance and normalization, the 23 unique cell cluster markers were identified across all cells in the six samples, each distinguished by unique colors. (B) The t-SNE plot visualizes the merging of three patients, with cells being uniformly distributed across all samples. This indicates that there are no notable batch effects within the HCC clusters. (C) Based on the marker genes present, the cells were classified into five distinct cell types: hepatocyte, myeloid cells, NK/T cells, cholangiocyte, and endothelial cells. (D) Distribution of tumor cells and normal cells. (E) Glycosylation activity score(G-score) in each cell was calculated using the AddModuleScore function. Cells were categorized into high G-score and low G-score subgroups using the median G-score. (F) Dot plot showing marker genes for various cell types. The size of the dots indicates the average proportion of cells expressing the target gene, while the color represents the average expression. (G) Distribution of high and low G-score groups in different cell subpopulations. (H) Heat map depicting the significant enrichment pathways between groups with high and low G-score values as revealed by GSVA. Blue color marks the enrichment pathway in the high G-score group, while green color indicates the pathway associated with the low G-score group.

Prognosis, clinical features and immune pattern of HCC patients closely related to glycosylation levels

To explore the effect of glycosylation levels on the prognosis and pathologic features of HCC patients, we used an unsupervised machine learning approach to group the 343 HCC patients in TCGA, using the differential genes obtained in Table S2 as a reference, with the best effect at the threshold value K =2 (Fig. 2A). The distribution of the cumulative distribution function (CDF) values across different groups in unsupervised clustering analysis is illustrated in Fig. 2B. We analyzed the differential genes between group 1 and group 2 using limma R package, and the results showed that the genes highly expressed in group 1 were TMSB10, MYBL2, CA9 and EPCAM (Fig. 2C).

Figure 2 Unsupervised clustering analysis.

(A) Consensus matrix for the best classification threshold (k = 2) in the TCGA cohort. (B) The distribution of the CDF values across different groups in the unsupervised clustering analysis. (C) Volcano plot showing differential expression genes between cluster 1 and cluster 2, with red representing genes highly expressed in cluster 1 and purple representing genes highly expressed in cluster 2 (D) K-M survival analysis for cluster 1 and cluster 2. The violin plot shows a difference in stromal scores (E) and immune scores (F) between cluster 1 and cluster 2, as determined by estimate algorithm. ∗, p < 0.05; ∗∗ P < 0.01.

Additionally, a K-M survival analysis conducted for prognosis demonstrated superior outcomes for cluster 1 compared with cluster 2 (P < 0.001) (Fig. 2D). The distribution and differences in clinical information between clusters 1 and 2 are depicted in Fig. S6, which includes aspects such as cancer type, Child-Pugh classification, gender, history, grading, medical history, TNM staging, overall staging, and vascular invasion. Moreover, using the ESTIMATE method, we identified differences in the stromal score (Fig. 2E) and immune score (Fig. 2F) between clusters 1 and 2. These results suggest that the prognosis, clinical features and immune pattern of HCC patients are closely related to the level of glycosylation.

Identification of glycosylation-associated prognostic hub genes by ssGSEA and WGCNA

In order to further identify the prognostic characteristics of the found glycosylation-related genes, using GRGs as a reference, we calculated the glycosylation scores of each HCC patient in the TCGA cohort using ssGSEA, and patients were categorized into high- and low-glycosylation groups according to the median glycosylation score (different from the G-score mentioned above). Our K-M survival analysis revealed that the group with high glycosylation had a less favorable prognosis than the group with low glycosylation (P = 0.005) (Fig. 3A), suggesting that glycosylation might be a risk factor for HCC, validating our above findings. To narrow down GRGs potentially closely associated with HCC prognosis, we utilized WGCNA to construct a gene co-expression network for patients with HCC. (Figs. S5A–S5B). Furthermore, we performed gene clustering between different modules using the dynamic tree cutting and dynamic hybrid methods (Fig. 3B). Figure 3C shows the correlation between phenotypic traits and the expression of different modules, displaying the correlation coefficients (P-values). The MEturquoise and MEgrey modules identified by WGCNA exhibited an association with glycosylation and survival status and were thus selected for further examination in subsequent stages.

Figure 3 SsGSEA and WGCNA.

(A) A glycosylation score was computed for each HCC patient in the TCGA cohort through ssGSEA. We performed a K-M survival analysis on high-glycosylation group and low-glycosylation group. (B) The graph depicts the gene clustering of different modules. (C) Heat map showing the relationship and importance between clinical characteristics and module eigengenes. In parenthesis, correlation coefficients and p-values are displayed. The modules MEturquoise and MEgrey highlighted in red show strong associations with glycosylation and survival status. Fultime = Follow-up time, fustat = Follow-up status.

Screening of glycosylation-related signatures for predicting HCC survival status using machine learning methods

MEturquoise and MEgrey modules, which are related to glycosylation and patient prognosis, included 767 genes. (Table S3). To further investigate the connection between these genes and the prognosis of patients with HCC, we referred to a previous study to create a prognostic marker for GRGs. (Wang et al., 2022). After performing univariate regression analysis, 223 genes with P-values lower than 0.001 were identified. Subsequently, we used LASSO Cox regression analysis to filter out the most specific features (Figs. 4A–4B). Ultimately, under the optimal regularization setting, we constructed a model with six features using multivariate Cox regression, the coefficients of which are shown in Fig. 4C. Among them, SPP1, SAPCD2, and S100A9 were positively correlated, while SOCS2, RAMP3, and CSAD were negatively correlated. We divided the HCC-TCGA cohort into a training cohort(n = 172) and a validation cohort (n = 171) and then scored and classified them into NHGRM_low and NHGRM_high. We utilized the log-rank test to compare differences in OS between subgroups, and found that in both cohorts, the OS of the NHGRM_high group was shorter than that of the NHGRM_low group (Figs. 4D–4E). Moreover, the ROC curves showed good prognostic accuracy in the training cohort (AUC:0.824 for 1-year, 0.817 for 3-year, 0.818 for 5-year) (Fig. 4F) and validation cohort (AUC:0.816 for 1-year, 0.669 for 3-year, 0.653 for 5-year) (Fig. 4G). In this instance, the AUC values of the model in the training cohort varied between 0.81 and 0.82, indicating that it has a strong chance of accurately predicting the prognosis of HCC patients. Similar outcomes were observed in the validation cohort.

Figure 4 Establishment and confirmation of a model involving 6 Glycosylation-related signatures within the TCGA-HCC cohort.

(A) 10-fold cross-validation was used to adjust the parameter selection. (B) LASSO coefficients are depicted on the Y-axis, while the X-axis represents -log(lambda). (C) The figure presents the coefficients of the 6 GRGs selected using LASSO regression analysis, with blue representing positively correlated genes and red representing negatively correlated genes. KM method analysis of OS differences between different subgroups in the TCGA-HCC training cohort (D) and validation cohort (E) was assessed by log-ranking test. (F) The time-dependent ROC curve showcases the prognostic accuracy of the NHGRM for 1-, 3-, and 5-year OS in the training set, with respective AUC values of 0.824, 0.817, and 0.818. (G) AUC values for the risk score’s prediction of in the validation cohort were shown.

Development and validation of prognostic nomogram

To better predict the prognosis of patients with NAFLD-associated HCC, we constructed and validated a prognostic nomogram combining risk scores and traditional clinical features. TNM system (tumor-node-metastasis) staging, “Tumor (T)” describes the size and extent of the primary tumor, “Node (N)” reflects the presence and extent of spread to nearby lymph nodes, and “Metastasis (M)” indicates whether the cancer has metastasized to other organs. The univariate Cox regression analysis, represented in a forest plot, indicated that ‘T’ (hazard ratio (HR) =1.699, 95% confidence interval (CI):1.365−2.115, P <0.001), ‘history’ (HR =1.459, 95% CI [1.113–1.913], P = 0.006), and ‘Risk Score’ (HR =1.158, 95% CI [1.114–1.205], P <0.001) are prognostic factors impacting the OS of HCC patients (Fig. 5A). The multivariate Cox regression analysis further suggested that ‘cancer type’ (HR =1.883, 95% CI [1.095–3.240], P = 0.022), ‘history’ (HR =2.039, 95% CI [1.097–3.769], P = 0.024), and ‘Risk Score’ (HR =1.173, 95% CI [1.117–1.232], P <0.001) affect the OS of HCC patients (Fig. 5B). A nomogram was subsequently constructed to predict the 1-, 3-, and 5-year survival rates of patients with HCC in the TCGA cohort (Fig. 5C). DCA was conducted to compare the clinical utility of each characteristic and nomogram based on the threshold probability. These findings indicated that augmenting the risk score with clinical variables has the potential to enhance the accuracy of survival prediction (Fig. 5D). The nomogram, ‘Risk’, and ‘Stage’ emerged as the superior predictors. As shown in Fig. 5E, the AUC values of the nomogram and glycosylation-based risk grouping in predicting patient prognosis were 0.834 and 0.826, respectively, which were markedly superior to those of other clinical features. The calibration curve revealed consistency between the 1-, 3-, and 5-year survival rates predicted by the nomogram and actual survival rates (Fig. 5F).

Figure 5 Construction and assessment of a prognostic nomogram.

(A) The forest plot illustrates the outcomes of the univariate Cox regression, indicating that ‘T’, ‘history’, and ‘risk score’ are significant prognostic factors that impact the OS of patients with HCC. (B) Multivariate Cox regression suggests that ‘cancer type’, ‘history’, and ‘risk score’ are prognostic factors affecting the OS of HCC patients. (C) A nomogram was constructed to estimate the survival rates at 1, 3, and 5 years for HCC patients within the TCGA cohort. The red line provides an illustrative example of prognostic prediction. (D) DCA was performed to evaluate the clinical utility of each feature and nomogram based on threshold probability. The lines higher on the DCA curve indicate greater net benefits. Among these, the nomogram, ‘Risk’ and ‘Stage’ demonstrated superior effects. (E) The nomogram plot and the glycosylation-based risk grouping yielded significantly higher AUC values in predicting patient prognosis than those of other clinical features. (F) The calibration curve demonstrates the consistency between the 1-, 3-, and 5-year survival rates forecasted by the nomogram and the actual survival rates. TNM system (tumor-node-metastasis) staging, “Tumor (T)” describes the size and extent of the primary tumor, “Node (N)” reflects the presence and extent of spread to nearby lymph nodes, and “Metastasis (M)” indicates whether the cancer has metastasized to other parts of the body.

Different clinical features in different NHGRM groups

To explore the differences between the different NHGRM groups, we assessed seven clinical characteristics. Vascular invasion status was categorized into macro, micro, and none, with significant differences observed across these categories (P = 0.0065 for macro vs. microvascular, P = 0.0005 for macro vs. none, and P = 0.0026 for microvascular vs. none) (Fig. 6A). The ‘T’ feature revealed statistically significant disparities in risk scores across the T1 and T2/3/4 (P = 3.9e−06 for T1 vs. T2, P = 2.4e−07 for T1 vs. T3, and P = 0.00029 for T1 vs. T4) (Fig. 6B). The ‘stage’ feature also displayed notable differences in risk scores between stages I and II/III (P = 1.4e−05 for stage I vs. stage II, P = 8.7e−09 for stage I vs. stage III) (Fig. 6C). The history of patients also showed a significant variation in risk scores (P = 0.0018) (Fig. 6D). The grade feature demonstrated significant differentiation between G1 and other grades (P = 0.00012 for G1 vs. G2, P = 1e−08 for G1 vs. G3, and P = 0.0011 for G1 vs. G4), with a noticeable difference between G2 and G3 (P = 0.0013) (Fig. 6E). However, there was no statistically significant difference in the gender features (P = 0.097) (Fig. 6F). Finally, cancer types represented by tumor-free and with tumor exhibited a significant discrepancy in HR risks (P = 0.03) (Fig. 6G).

Figure 6 Boxplots represent the distribution and statistical significance of seven distinct features in the data set, including vascular (A), T (B), stage (C), history (D), grade (E), gender (F) and cancer type (G).

TNM system (tumor-node-metastasis) staging, “Tumor (T)” describes the size and extent of the primary tumor.

Predicting the composition and immune profile of immune cells in the tumor microenvironment in different NHGRM groups

To explore the mechanisms underlying the prognostic differences between patients with different risk scores, we calculated the proportional distribution of 22 immune cell types in the TCGA-LIHC dataset using the CIBERSORT algorithm (Fig. 7A). Furthermore, we explored the effect of the level of infiltration of 22 cell types on the prognosis of patients, and we explored the effect of infiltration levels of 22 cell types on patient prognosis, and we found that HCC patients with higher levels of NK-activated cell infiltration had a better prognosis, however, HCC patients with higher levels of macrophage M1 infiltration had a worse prognosis, and there were no significant differences between the other cells (Fig. 7B). Interestingly, the levels of NK-activated cells, T cells CD8, and mast-activated cell infiltration were higher in the NHGRM_low group than in the NHGRM_high group. However, the levels of macrophage M0 infiltration were higher in the NHGRM_high group (Fig. 7C). Moreover, we explored the expression levels of immune checkpoints in different NHGRM groups. TNFRSF18, IFNG, PDCD1, LGALS9, LDHA, IL12A, CD80, YTHDF1, TNFRSF9, HAVCR2, TNFSF9, CD86, VTCN1, CTLA4, TNFRSF4, ICOS, and TNFSF4 were less expressed in the low-NHGRM group compared to the NHGRM_high group. However, ICOSLG, SIGLEC15, PVR, IL23A, FGL1, JAK2 and CD274 were expressed at higher levels in the low-NHGRM group (Fig. 7D).

Figure 7 Immune infiltration and immune checkpoints in different NHGRM groups.

Distribution of 22 immune cell infiltrations in HCC patients in TCGA. (B) Prognostic impact of different immune cell expression levels in TCGA patients, with macrophage M1 in the left panel and activated NK cells in the right panel. (C) Differences in the proportion of immune cell infiltration between groups. (D) Expression of immune checkpoints in different NHGRM groups. Genes marked in red are significantly different genes in NHGRM-high group vs NHGRM-low group, ∗ P < 0.05; ∗∗P < 0.01; ∗∗∗P < 0.001.

Validation of the NAFLD-HCC glycogene risk model in vivo and in vitro

We further validated our prognostic models using the external HCC dataset, GSE54236. Survival outcomes between the NHGRM_low and NHGRM_high groups were found to be significantly different in the K-M survival analysis conducted using risk stratification (P = 0.036) (Fig. 8A). This distinction validates the robustness of our risk models, demonstrating that NHGRM_high patients experienced poorer survival outcomes. Moreover, the ROC curves showed good prognostic accuracy in the GSE54236 HCC cohort (AUC:0.811 for 1-year, 0.725 for 2-year, 0.612 for 3-year) (Fig. 8B).

Figure 8 Validation of external dataset GSE54236 and experimental validation.

(A) K-M survival curves derived from the risk stratification, revealing a significant survival disparity in the different NHGRM subgroups ( P < 0.05). (B) Analysis of the prognostic efficacy of the model was conducted using a ROC curve. The model exhibited its highest predictive performance for one-year OS with an AUC of 0.811, while the AUCs for two-year and three-year OS were 0.725 and 0.612, respectively. (C–D) Pathological images of liver tissues from mice fed with mixed grains. (E–F) HE stains of liver tissues from mice fed with a high-fat diet. (G–L) The box plots display the expression differences in six signatures (RAMP3, S100A9, SAPCD2, SOCS2, SPP1, and CSAD) between the normal control (NC) group and the NAFLD group. ∗∗, P < 0.01; ∗∗∗, P < 0.001.

To further explore the NAFLD-HCC glycogene risk model (NHGRM) signatures in the progression of NAFLD-HCC, we constructed a high-fat diet-induced NAFLD model in mice by feeding for 16 weeks (Fig. S4A). Mouse liver appearance (Fig. S4B), body weight (Fig. S4C), liver weight (Fig. S4D), fat weight (Fig. S4E), and serum triglycerides (Fig. S4F) were displayed at a higher level in the high-fat diet group compared with the normal diet group at 16 weeks. HE staining of liver tissues showed the presence of steatosis and inflammatory infiltration in the livers of the NAFLD group (Figs. 8C–8F). These results indicated that the experimental mice successfully modeled NAFLD. Furthermore, we examined the mRNA levels of six signatures in mouse liver and showed that Ramp3, S100a9, Sapcd2, Spp1 and Csad were exhibited at higher levels in the NAFLD group compared to NC group (Figs. 8G–8I, 8K, 8L). Socs2 expression was higher in the NC group (Fig. 8J).

Subsequently, two NAFLD datasets were identified for further external validation. The GSE48452 dataset was divided into four groups: control, NASH, healthy obese, and steatosis. Using the t-test, we observed differences in the expression levels of five NHGRM signatures(CSAD, RAMP3, S100A9, SOCS2, and SPP1, SAPCD2 not detected) across these groups, as illustrated in Figs. 9A–9E. Notably, the expression of SOCS2 in the NASH group was downregulated compared to that in the control or healthy obese groups, while the NASH group exhibited higher SPP1 expression (P = 0.0023 for NASH vs. Control, P = 0.0059 for NASH vs. healthy obese). We also conducted correlation analyses to clarify the relationships between these NHGRM signatures and several physiological variables in the dataset, including Body Mass Index (BMI), adiponectin, leptin, lar, and NAS scores. For instance, CSAD showed a positive correlation with lar, leptin, and BMI, but was negatively correlated with adiponectin. (Fig. 9F). The remaining dataset, GSE89632, was divided into three groups: simple steatosis, NASH, and healthy controls. Through the Kruskal-Walli’s test, we also discovered statistically significant variations in the expression levels of these NHGRM signatures across these groups. (P = 5.6e−07 for CSAD, P = 0.024 for RAMP3, P = 1.7e−05 for S100A9, P = 6.6e−10 for SOCS2, and P = 0.00023 for SPP1) (Figs. 9G–9K). Figure 9L illustrates the relationships between NHGRM signatures and various physiological variables including BMI, cholesterol, triglycerides, transaminases, waist circumference, hemoglobin, and age. For example, CASD was positively correlated with total cholesterol and transaminases, whereas SOCS2 was negatively correlated with alkaline phosphatase, BMI, and waist circumference.

Figure 9 Validation was performed on the external datasets GSE48452 and GSE89632.

(A–E) T-test analysis of box plots showing the variances in expression levels of five GRGs within the GSE48452 dataset group. (F) Heat map from the GSE48452 dataset illustrating the relationship between glycation genes and several physiological parameters. Dark blue in the graph indicates positive correlations, dark red indicates negative correlations, and entries with stars indicate statistical significance. ∗, P < 0.05; ∗∗, P < 0.01; ∗∗∗, P < 0.001. (G–K) Box plots show the comparison of expression levels of five GRGs among three groups in the GSE89632 dataset. (L) Heat map demonstrating the correlation between glycation genes and various physiological variables within the GSE89632 dataset. Colors and asterisks have the same meaning as above.

Furthermore, we examined the protein expression levels of NHGRM signatures in the livers of patients with HCC. Immunohistochemical staining images from the Human Protein Atlas database showed that CSAD and SAPCD2 were highly expressed in patients with tumors, compared to normal liver, while SOCS2 was lowly expressed in patients with HCC (Figs. 10A–10C). Since there is no data of OPN (SPP1 gene), S100A9, and RAMP3 in HPA database, we detected their expression in tumor tissues and paracancerous tissues using immunofluorescence staining in shuguang cohort, and the results showed that OPN, S100A9, RAMP3 were highly expressed in tumor tissues compared with paracancerous tissues (Figs. 10D–10F). We also detected the expression of characterized genes in the single-cell dataset (Fig. S3). It consistent with the trend of protein levels in liver tissue (Fig. S3C). Specifically, CASD was highly expressed in myeloid cells, while RAMP3 was highly expressed in endothelial cells (Fig. S3D). In summary, we explored the expression of NHGRM signatures in one tumor dataset and two NAFLD datasets, NAFLD animal models, and human tissue samples.

Figure 10 Protein expression levels of NHGRM signatures in the livers of patients with HCC.

(A–C) Immunohistochemical data from the Human Protein Atlas database revealed elevated expression of CSAD and SAPCD2 in tumor patients relative to normal liver, and diminished SOCS2 expression in HCC cases. (D–F) Immunofluorescence staining results indicated that the expression of OPN, S100A9, and RAMP3 was higher in tumor tissues than in paracancerous tissues. Blue color represents dapi and red color represents target protein.

Discussion

Alterations in glycosylation patterns have been observed in hepatic pathologies and are implicated in tumorigenesis, progression, and metastasis, suggesting that the modulation of glycosylation impacts a range of proteins integral to the pathogenesis of NAFLD (Ramachandran et al., 2022). Moreover, increased core fucosylation, branching, and sialylation of glycans have been identified in individuals diagnosed with NASH and HCC (Ramachandran et al., 2022). The functional contributions of glycosyltransferases, along with other biochemical pathways, such as phosphoric acid oxidation, FASN-mediated lipid biosynthesis, and glycolysis, known as the “Warburg effect,” are deemed critical to the progression of these hepatic diseases (Gabbia, Cannella & De Martin, 2021; Che et al., 2019; Zhan, Su & An, 2016). Therefore, investigating the specific mechanism of GRGs in NAFLD and NAFLD-associated HCC will provide new clues for diagnosis, prognosis, and treatment.

Single-cell sequencing technology can detect tumor microenvironments and extract gene expression profiles from HCC cells, which is crucial for early detection and treatment targeting of NAFLD-related HCC (Hedlund & Deng, 2018; Zou et al., 2023). However, it also poses challenges, such as the initial isolation and culturing of individual cells. Many analytical techniques can inadvertently damage cells during the process, contributing to inaccurate results. Therefore, in our study, we integrated bulk RNA-seq and scRNA data to fully utilize their respective advantages. We investigated GRG expression patterns using single cell sequencing datasets. We initially identified a multitude of cell subpopulations within HCC and observed that GRG activity varied among these cell lineages. According to the GSVA algorithm, our analysis revealed a strong enrichment of the high G-score subgroup in oxidative phosphorylation, adipogenesis, fatty acid metabolism, and glycolysis signaling pathways, all of which warrant further rigorous investigation. To determine the GRGs most relevant to NAFLD-HCC prognosis, we applied unsupervised clustering analysis.

To establish a prognostic model for TCGA liver cancer, we used a univariate-lasso-multivariate Cox regression model. Using LASSO-Cox regression analysis, we further reduced overfitting and identified six signatures: SPP1, SOCS2, SAPCD2, S100A9, RAMP3, and CSAD. We constructed the optimal glycosylation-related prognostic features and validated them using GSE54236. A scoring formula was derived by utilizing LASSO coefficients and GRG expression levels, resulting in a risk score. Based on this, patients with HCC were categorized into the NHGRM_low and NHGRM_high groups. Remarkably, the NHGRM_high group exhibited worse prognosis, irrespective of clinical parameters. After the prognostic features in both the training and validation cohorts demonstrated good predictive capabilities, we investigated the underlying mechanisms. As expected, disparities in immune infiltration and immune checkpoint levels were observed between the NHGRM_low and NHGRM_high groups, which could potentially lead to tumor heterogeneity.

Glycosylation is considered to be the most complex post-translational modification involved in cell signaling and communication, tumor cell dissociation and invasion, cell–matrix interactions, tumor angiogenesis, immune regulation, and metastasis formation in tumor development (Pinho & Reis, 2015). Core petaloid glycosylated alpha-fetoprotein (AFP-L3) can be a sensitive and specific circulating biomarker for the early diagnosis of HCC, suggesting that glycosylation-associated genes play an important role in the early diagnosis of tumors. Therefore, in this study, we focused on the expression of key genes closely related to glycosylation during the progression of NAFLD to HCC, as well as the expression patterns in different cells, to shed light on the mechanism of NAFLD progression to HCC.

OPN is an extracellular glycosylated phosphoprotein that promotes cell adhesion by interacting with several integrin receptors (Oyama et al., 2018). It has been demonstrated that OPN O-glycosylation self-regulates its biological activity and influences its phosphorylation status (Kariya et al., 2014). However, extracellular OPN promotes obesity and regulates lipid synthesis, which in turn leads to hepatic steatosis (Nomiyama et al., 2007; Nuñez Garcia et al., 2017). In HCC, OPN is also highly expressed, which is closely related to its mode of glycosylation and the mechanism of promoting cell adhesion for tumor metastasis, which is consistent with our study. Not only that, recent scRNA-Seq analysis of human cirrhotic (Ramachandran et al., 2019) and HCC (Zhang et al., 2019) livers showed increased expression of SPP1 in inflammatory cell subsets (i.e., MFs, T cells, dendritic cells, and NK cells). In our results, SPP1 was highly expressed in HCC patients and was increased in NK and Myeloid cells, suggesting that it is closely associated with the level of inflammation, and chronic inflammation leads to the progression of cirrhosis and HCC. Suppressor of cytokine signaling 2 is a protein that in humans is encoded by the gene (Elliott & Johnston, 2004). There are few reports between SOCS2 and glycosylation, so the link between them is not clear for the time being. However, previous studies have shown that SOCS2, as an anti-inflammatory substance, can inhibit inflammation and apoptosis during the progression of NASH by activating the activation of NF- κB, and its expression in macrophages was confirmed. There are few reports between SOCS2 and glycosylation, so the link between them is not clear for the time being. However, previous studies have shown that SOCS2, as an anti-inflammatory substance, can inhibit inflammation and apoptosis during the progression of NASH by activating the activation of NF- κB, and its expression in macrophages was confirmed (Li et al., 2021). SOCS2 plays a protective role in tumorigenesis and may be associated with its activation of specific cell death modalities such as Ferroptosis (Chen et al., 2023). A positive correlation between survival time and SOCS2 expression level in HCC tumor patients was also found in our study. This suggests the need to go further to explore the effects of possible glycosylation modifications on SOCS2 function and activity. SAPCD2, as an oncogene, can promote cell proliferation and tumor development, potentially affecting pathways such as PI3K/Akt, MAPK, and Hippo, although its direct effect on immune checkpoints is not yet clear (Zhang et al., 2022). The four glycosylation sites at the extracellular N-terminal end of RAMP3 play important physiological roles in binding to other proteins (Flahaut et al., 2003). Interestingly, a genome-wide study found an association between low-grade fat accumulation and rs10859525 and rs1294908, which are located upstream of SOCS2 and RAMP3, respectively, suggesting that SOCS2 and RAMP3 may serve as predictors of NAFLD disease progression (Di Stefano et al., 2015). It is well known that S100A9 is a regulator of bone marrow-derived immune cells, and it has been reported that glycosylation-dependent interactions between S100A9 and the complex between CD69 are required for regulatory T cell differentiation (Lin et al., 2015). CSAD, a protein-coding gene, plays a role in regulating taurine metabolism and is associated with diseases such as Stiff-Person Syndrome and various autoimmune disorders (Sköldberg et al., 2004). Some studies have reported that overexpression of CSAD improves fatty liver, but it is less commonly reported in disease progression to e.g., NASH and HCC (Tan et al., 2022). In conclusion, all six-signature required for our model have some ability to predict disease, but the causal link between their expression and disease still needs to be further explored.

Research may also be able to identify and screen for potential beneficiaries of immunotherapy because glycosylation, which exhibits higher sensitivity to immunotherapy, may have a substantial impact on the metabolic pathways associated with HCC, including bile acid metabolism and fatty acid metabolism (Shi et al., 2022). With the rapid development of immunotherapy, significant advancements have been made in the treatment of HCC. Atezolizumab (anti-PD-L1) in combination with Bevacizumab (anti-VEGF-A) is the only guideline-recommended preferred regimen for first-line treatment (Finn et al., 2020). Anti-PD1 drugs are used as treatment choices in various regions after the use of anti-angiogenesis tyrosine kinase inhibitors (TKIs) (Sangro et al., 2021). Moreover, extensive research has demonstrated that immune checkpoint inhibitors (ICPIs) have significant effects on HCC therapy, with over 200 types of immunotherapeutic drugs currently undergoing clinical trials (Zheng et al., 2021). The study carried out by Peng et al. (2021) identified and validated seven immune-related genes (IRGs) associated with HCC. These IRGs are strong independent prognostic factors for the survival of patients with liver cancer. They may reveal information on immune cell infiltration into the TME and the state of immune suppression and may predict how well HCC patients will respond to immunotherapy (Peng et al., 2021). In our study, we found that some of the targets PDCD1, CTLA4, etc., which are already pharmacologic, were more highly expressed in the high-risk group. Our study also revealed a correlation between glycosylation levels and immune cell infiltration. A meta-study that included 26 studies noted that in patients with HCC, high NK cell levels were associated with better overall survival and disease-free survival (Xue et al., 2022). Cytotoxic T-lymphocytes (CTL) consistently possess CD8 surface antigen and play a role in anti-tumor immune responses. Studies have shown that high expression of CD8+ TIL is associated with a favorable prognosis in a variety of tumors, including HCC (Sun et al., 2015; Zhao et al., 2019). We found that the low-risk group had higher levels of infiltration of NK-activated cells, CD8 T cells, and mast-activated cells, while the high-risk group had higher levels of macrophages of the M1 type.

Although this study identified signatures associated with glycosylation that can be used to predict prognosis in patients with NAFLD-related HCC, it is not without limitations. Because of the heterogeneity of HCC and the fact that our features were developed and validated in a relatively small sample size cohort, validated only in our own small clinical cohort, it is important to validate the predictive power of the model in a large multicenter cohort before applying it to clinical practice. Our animal model did not develop tumors because high-fat diet-induced tumor models in mice take longer (more than a year) and have lower tumor incidence. Although we detected genes associated with the characteristics of patients with NAFLD-HCC, however, more in-depth exploration through further experimental work is still needed to understand the potential molecular pathways from NAFLD to HCC.

Conclusions

In this study, we constructed a prognostic model that can be used to predict the prognosis associated with NAFLD-HCC based on glycosylation levels and validated it at the animal and clinical levels, providing new perspectives on the role of glycosylation in tumorigenesis and development.

Supplemental Information

Supplemental Information 1 Data processing flow and visualization

R software

Supplemental Information 2 Flowchart of this article

Our aim was to identify glycosylation hub genes with prognostic significance in NAFLD-associated HCC by a combinatorial algorithm, to build a prognostic prediction model, and to validate its performance and biological significance. Through several steps: 1) scoring all cells through 636 GRGs (Table S1) using the AddModuleScore function Seurat R package to identify differential genes in the scRNA-seq dataset; 2) based on these differentially expressed genes, using an unsupervised clustering algorithm to categorize patients with hepatocellular carcinoma in the TCGA database into two groups, in order to to verify that glycosylation levels can affect patients’ prognosis; 3) Using GRGs as a reference, the ssGSEA algorithm was used to calculate the glycosylation levels of each patient in TCGA-LIHC, and then WGCNA was used to screen out the modules that were highly correlated with the prognosis and glycosylation levels. 4) Then use univariate cox-lasso-Multivariate cox regression to screen the features among modules and build models that can predict the prognosis of HCC patients. 5) Divide the TCGA into a training set and an internal validation set to validate the performance of the models. 6) Construct a column-line graph. 7) Explore the clinical features and immune infiltration of different risk models. 8) In the external validation set (1 hepatocellular carcinoma and 2 NAFLD) sets, animal models and human liver tissues to validate the biological significance of the model features.

Supplemental Information 3 Detailed presentation of the markers used for cell categorization

Dot plot showing marker genes for the five cell types defined in the article. The size of the dots indicates the average proportion of cells expressing the target gene, while the color represents the average expression. (B) Dot plot indicating marker genes for 23 distinct clusters. The meaning of the size and color of the dots in the figure is the same as above.

Supplemental Information 4 Expression of each marker in the single-cell dataset

(A) t-SNE plots show the distribution in different cell subpopulations. Dot plots show the expression of selected genes (SPP1, SOCS2, SAPCD2, S100A9, RAMP3, and CSAD) in different clusters(B), in normal and tumor tissues(C), and in different cell subpopulations(D).

Supplemental Information 5 Induction of mice model of nonalcoholic fatty liver

(A) Flow chart of high-fat diet-induced NAFLD in mice. Liver appearance (B), body weight (C), liver weight (D), fat weight (E), and serum triglycerides (F) at 16 weeks in high-fat and normal diet-fed mice.

Supplemental Information 6 WGCNA

Utilizing WGCNA, gene co-expression networks for HCC patients were constructed. The prevalence and tendencies of scale-free topological model fit(A) and average connectivity (B).

Supplemental Information 7 The distribution and differences in clinical information between two clusters

The bar plot displays the distribution and differences in clinical information between cluster 1 and cluster 2, including cancer type, Child-Pugh grade, gender, histological grade, medical history, TNM staging, overall stage, and vascular invasion.

Supplemental Information 8 Raw data and code of animal experiments

PCR rawdata and the scripts of figure 10 and the MIQE checklist of PCR.

Supplemental Information 9 Author checklist

Supplemental Information 10 Appendix and integration of all supplementary tables

Supplemental Information 11 MIQE checklist of qPCR

List of Abbreviations

NAFLD non-alcoholic fatty liver disease

HCC hepatocellular carcinoma

GRGs glycosylation-related genes

WGCNA Weighted Gene Co-expression Network Analysis

NHGRM NAFLD-HCC GlycoGene Risk Model

NASH Non-alcoholic steatohepatitis

DEGs Differentially expressed genes

TCGA-LIHC The Cancer Genome Atlas - Liver Hepatocellular Carcinoma

LASSO Least Absolute Shrinkage and Selection Operator

scRNA-seq Single-Cell RNA Sequencing

GEO The Gene Expression Omnibus

FPKM fragments per kilobyte per million

TPM transcripts per million reads

MSigDB Molecular Signature Database

TME tumor microenvironment

QC quality control

PC principal components

t-SNE t-distributed Stochastic Neighbor Embedding

AUC area under the curve

GSVA Gene Set Variation Analysis

PAM partitioning around the median

ssGSEA single-sample gene set enrichment analysis

TOM topological overlap matrix

OS overall survival

K-M Kaplan–Meier

ROC receiver operating characteristic

DCA decision curve analysis

ICG immune checkpoint genes

HFD high-fat diet

C group control group

qPCR quantitative Polymerase Chain Reaction

qRT-PCR quantitative real-time PCR

H&E Hematoxylin and eosin

CDF Cumulative Distribution Function

T Tumor

N Node

M Metastasis

HR Hazard Ratio

CI Confidence Interval

BMI Body Mass Index

mTOR Mammalian Target of Rapamycin

TAMs tumor-associated macrophages

WD Western diet

Treg regulatory T cells

TKIs tyrosine kinase inhibitors

ICPIs immune checkpoint inhibitors

IRGs immune-related genes

ICIs immune checkpoint inhibitors

OPN Osteopontin

FFPE Formalin-fixed paraffin-embedded.

Additional Information and Declarations

Competing Interests

Author Contributions

Animal Ethics

Data Availability

The authors declare there are no competing interests.

Zhijia Zhou conceived and designed the experiments, performed the experiments, analyzed the data, prepared figures and/or tables, and approved the final draft.

Yanan Gao conceived and designed the experiments, analyzed the data, prepared figures and/or tables, and approved the final draft.

Longxin Deng conceived and designed the experiments, performed the experiments, analyzed the data, prepared figures and/or tables, and approved the final draft.

Xiaole Lu conceived and designed the experiments, analyzed the data, prepared figures and/or tables, and approved the final draft.

Yancheng Lai performed the experiments, prepared figures and/or tables, and approved the final draft.

Jieke Wu conceived and designed the experiments, prepared figures and/or tables, and approved the final draft.

Shaodong Chen conceived and designed the experiments, authored or reviewed drafts of the article, and approved the final draft.

Chengzhong Li conceived and designed the experiments, authored or reviewed drafts of the article, and approved the final draft.

Huiqing Liang conceived and designed the experiments, authored or reviewed drafts of the article, and approved the final draft.

The following information was supplied relating to ethical approvals (i.e., approving body and any reference numbers):

Experiments were conducted following established animal protocols and the guidelines approved by the animal experimental ethics committee of Xiamen University (approval No. XMULAC20200055).

The following information was supplied regarding data availability:

The data is available at NCBI GEO: GSE189175, GSE48452, GSE89632 and The Cancer Genome Atlas (TCGA; https://portal.gdc.cancer.gov/).

The raw measurements are available in the Supplemental Files.

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
