# Peer review of "Integrating single-cell and bulk sequencing data to identify glycosylation-based genes in non-alcoholic fatty liver disease-associated hepatocellular carcinoma"

_PeerJ, doi:10.7717/peerj.17002_

## Round 0.1 · original submission · Major Revisions

1. Please provide better resolution figures.
2. Please address all the comments from the reviewers.
3. Please revise the immune cells section with clear descriptions of the different subtypes of immune cells analyzed (CD4, CD8, and macrophages).
4. Please fix all the typographical errors.

**Language Note:** The Academic Editor has identified that the English language must be improved. PeerJ can provide language editing services - please contact us at [email protected] for pricing (be sure to provide your manuscript number and title). Alternatively, you should make your own arrangements to improve the language quality and provide details in your response letter. – PeerJ Staff

·

Basic reporting

The article is well written.

Experimental design

The experiments are well executed.

Validity of the findings

The results are well explained & the conclusions are well stated.

Additional comments

In the article titled “Integrating single-cell and bulk sequencing data to identify glycosylation-based genes in non-alcoholic fatty liver disease-associated hepatocellular carcinoma” the authors have developed NHGRM which has the potential to serve as an indicator for survival of patients suffering from HCC associated with NAFLD. The study is very interesting, and the work is well executed.

My concerns are related to the in vivo validation of the study and are as follows:

1. It would be good if the authors can perform some more tests to confirm the induction of NAFLD in the liver tissue of the mice.
2. Also please provide statistics of the level of induction of NAFLD in the liver tissue of the mice.

·

Basic reporting

Check line 61 for typographical errors.
Provide additional description for line 263 pertaining to figure 1.

Experimental design

No comment

Validity of the findings

Provide Jpeg or tiff files for the 11 figures files for verification (supplemental information)

Additional comments

I commend the authors for their extensive data set, compiled over many years of detailed fieldwork. In addition, the manuscript is clearly written in professional, unambiguous language. If there is a weakness, it is some typographical errors and missing descriptions (as I have noted above) as well as jpeg or tiff files for figures for verification which should be improved upon before Acceptance.

Reviewer 3 ·

Basic reporting

I understand that several terms and abbreviations used in the manuscript are directly derived from the algorithm package or are commonplace within the field, but a more comprehensive explanation of these terms would significantly enhance the reader's comprehension of the presented methodologies and findings. This is particularly crucial within the results section and the figures and their corresponding captions. For instance, on lines 308-309, the authors mention "We obtained non-gray modules (Figs. 4C -4D)". Could the authors clarify what "non-gray modules" entail? If this is the typo, it appears to possibly refer to “non-grey modules” that are gene sets associated with specific traits. It is imperative that the authors provide an explanatory context for this term in relation to the data. This is also applicable to figures. In Figure 4F, the x-axis includes "fultime" and "fustat," yet there is an absence of preceding explanation or definition for these terms within the text. In Figure 6, rather than simply utilizing 'T', 'N', and 'M', the authors should elucidate what these labels represent (e.g., 'T': tumor size). Moreover, in Figure 6C, the use of " *** " needs clarification. Additionally, certain plots should be handled more carefully. For instance, why is the legend for Figure 1E and 1G different, and what does "glycosylation feature" refer to? Similarly, Figures 2C and 2D raise questions due to the y-axis being designated as "Down/up-regulated gene in diffscRNA-seq2". I am not sure how gene could be numerical. Could the authors potentially be referring to gene expression levels? This pattern extends to several other subsections within the results and a multitude of figures as well. Also, there a number of typos in manuscript and please proof read.


In terms of the analytical pipeline, it should be elaborated with enhanced detailing. For instance, the application of "Gscore" (line 276 and line 277) as an analytical parameter needs a comprehensive explanation. What precisely does Gscore denote, how is its computation conducted, and is it same as the "AddModuleScore" in Seurat, or is it based on GSVA? Similarly, lines 324-326, the categorization of NHGRM_low and NHGRM_high is based on previous research. Could the authors provide reference? Additionally, in lines 313 to 314, MEturquoise and MEgrey modules are selected for subsequent examination, followed by line 317 mentioning the use of WGCNA to screen 767 genes. Are these 767 genes from the MEturquoise and MEgrey modules? Same for the later sections.

Experimental design

Regarding the single-cell RNAseq dataset sourced from Alvarez et al., 2022, it's worth noting that while the original paper identified eight primary clusters, only five cell types are defined within this study. This disparity raises questions about the reasons behind these differences. Moreover, the authors identified 193 genes, in line 155-157, and use these genes to construct clusters and determine immune subtypes. Should this list of genes be acquired through the differential gene analysis conducted between the high and low Gscore groups within the immune cell clusters that are annotated in single cell data?

Validity of the findings

On lines 290-291, the assertion is made that within the high Gscore group, genes with elevated expression also show a higher expression level in cluster 1 (Figure 3D). To substantiate this claim, I recommend conducting a statistical analysis, as the difference illustrated in the plot appears relatively subtle.

In terms of the six specific genes (SPP1, SOCS2, SAPCD2, S100A9, RAMP3, and CSAD) could the authors provide insights into how these genes are represented in the single-cell data? Have similar trends been observed when comparing tumor and non-tumor single cell data?

Reviewer 4 ·

Basic reporting

In the submitted manuscript by Gao et al, the authors present a negative relation between glycosylation-related gene expression and incidence of hepatocellular carcinoma associated with non-alcoholic fatty liver diseases by integrating a comprehensive collection of RNA-seq and scRNA-seq data from the literature. They discovered six glycosylation-related genes as related prognostic factors in NAFLD associated HCC with higher expression linked to worse outcomes. Despite the above, I do have some comments/concerns with the submitted manuscript:
- As the authors mentioned in the conclusion. It is hard to connect the mouse model results to humans' as the mouse model do not induce tumors in mice which is limited to interpret. Could the authors add more comments on this? And is it because the basic level of Glycosylation-related genes are high in some people that are more prone to have liver diseases or it is the liver diseases induce the high expression of the genes?
- The overall resolution for the figures is poor. Please upload more higher resolution figures.
- The authors should add more details when describing the results and add the methods briefly
- The English language should be improved especially the discussion and conclusion sections.
Figure 1:
- please move this figure to supplement
Figure 2:
- F, are there other populations detected (eg. B cells) as indicated in Figure 8. Please also provide figures showing which signature genes are used to annotate the different populations.
- G, please show the annotation color bar.
Figure 3:
Not clear in describing the methods and conclusion related to this Figure in the context.
- F. please remove it to supplement as it does not seem provide important information.
- H. Does stromal score the same as matrix scores? Please be consistent.
Figure 4:
- A. What are those green dots? Not indicated in the context. And please also change the axis label to Log2 (fold change “of what”)
- C-E. Please also remove to the supplement figures as these figures are more about methods.
Figure 7:
Authors are kind of lost in immune cell population annotation. Fibroblasts and endothelial cells are not immune cells. T cells can be separated into CD4 T cells and CD8 T cells in general while the authors found T cells and CD8 T cells. Does that mean CD4 T cells? How about macrophage? What do you mean by cytotoxic lymphocytes? How do you annotate those population? What markers do you use as T cells are also cytotoxic lymphocytes. Please revise this section.
Discussion:
The discussion on CD4 T cells and B cells are not clear and do not make sense. Please revise and reorganize the sentence.
Typo or grammar mistakes:
line 394, was fed with
Line 426 such as
Line 478: with many subgroups?
Line 482 of immunotherapy, because
Line 487: using ani-PDL1 antibodies and?

Experimental design

no comments

Validity of the findings

no comments

---

## Round 0.2 · accepted · Accept

Thank you for addressing all the reviewer's comments.

·

Basic reporting

No comment

Experimental design

No comment

Validity of the findings

No comment

Additional comments

The authors have satisfactorily answered all my questions and concerns.

·

Basic reporting

All corrections have been addressed

Experimental design

All corrections have been addressed

Validity of the findings

All corrections have been addressed

Reviewer 3 ·

Basic reporting

No comment

Experimental design

No comment

Validity of the findings

No comment

Reviewer 4 ·

Basic reporting

I appreciate the authors' meticulous point-to-point response to my previous comments. It is evident that considerable effort has been invested in addressing the concerns raised. Overall, the paper is ready for publication.

Experimental design

N/A

Validity of the findings

N/A

Additional comments

N/A